# Current practices of management of maternal and congenital Cytomegalovirus infection during pregnancy after a maternal primary infection occurring in first trimester of pregnancy: Systematic review

Claire Périllaud-Dubois[1,2,3]*, Drifa Belhadi[2,4], Cédric Laouénan[2,4], Laurent Mandelbrot[2,3,5], Olivier Picone[2,3,5], Christelle Vauloup-Fellous[3,6,7]

1 Département de Virologie, AP-HP.Sorbonne Université, Hôpital Saint-Antoine, Paris, France, 2 Université de Paris, INSERM UMR1137, IAME, Paris, France, 3 GRIG, Groupe de Recherche sur les Infections pendant la Grossesse, Paris, France, 4 Department of Epidemiology, Biostatistic and Clinical Research, AP-HP, Hôpital Bichat, Paris, France, 5 AP-HP, Hôpital Louis Mourier, Service de Gynécologie-Obstétrique, Colombes, France, 6 Laboratoire de Virologie, AP-HP.Université Paris-Saclay, Hôpital Paul Brousse, Villejuif, France, 7 Université Paris-Saclay, INSERM U1193, Villejuif, France

* claire.perillaud-dubois@inserm.fr

## Abstract

### Introduction

Congenital CMV infection is the first worldwide cause of congenital viral infection but systematic screening of pregnant women and newborns for CMV is still debated in many countries.

### Objectives

This systematic review aims to provide the state of the art on current practices concerning management of maternal and congenital CMV infection during pregnancy, after maternal primary infection (PI) in first trimester of pregnancy.

### Data sources

Electronically searches on databases and hand searches in grey literature.

### Study eligibility criteria and participants

Primary outcome was listing biological, imaging, and therapeutic management interventions in two distinct populations: population 1 are pregnant women with PI, before or without amniocentesis; population 2 are pregnant women with congenitally infected fetuses (after positive amniocentesis). Secondary outcome was pregnancy outcome in population 2.

### Results

Out of 4,134 studies identified, a total of 31 studies were analyzed, with 3,325 pregnant women in population 1 and 1,021 pregnant women in population 2, from 7 countries

**Data Availability Statement:** All relevant data are within the manuscript and its Supporting Information files.

**Funding:** The authors received no specific funding for this work.

**Competing interests:** The authors have declared that no competing interests exist.

**Abbreviations:** CMV, Cytomegalovirus; CENTRAL, Cochrane Central Register of Controlled Trials; cCMV, congenital CMV infection; HIG, hyperimmunoglobuli; IQR, inter quartile range; IUFD, intra uterine fetal death; MeSH, Relevant medical subject heading; MRI, magnetic resonance imaging; NOS, Newcastle-Ottawa scale; PI, primary infection; PICOS, Population–Interventions–Comparators–Outcomes–Studies; PRISMA, Preferred Reporting Items for Systematic Review and Meta-Analysis; RCT, randomized controlled trial; RoB, risk of bias; TOP, termination of pregnancy; US, ultrasound.

(Belgium, France, Germany, Israel, Italy, Spain and USA). In population 1, ultrasound (US) examination frequency was 0.75/month, amniocentesis in 82% cases, maternal viremia in 14% and preventive treatment with hyperimmune globulins (HIG) or valaciclovir in respectively 14% and 4% women. In population 2, US examination frequency was 1.5/month, magnetic resonance imaging (MRI) in 44% cases at 32 weeks gestation (WG), fetal blood sampling (FBS) in 24% at 28 WG, and curative treatment with HIG or valaciclovir in respectively 9% and 8% patients.

## Conclusions

This systematic review illustrates management of maternal and congenital CMV during pregnancy in published and non-published literature, in absence of international consensus.

## Systematic review registration

PROSPERO CRD42019124342

## Introduction

Cytomegalovirus (CMV) is the first worldwide cause of congenital viral infection and its prevalence is currently estimated between 0.5 and 1% of all live births. Congenital CMV (cCMV) is a major cause of sensorineural hearing loss and mental retardation [1–3]. Vertical CMV transmission from mother to fetus can occur after primary (PI) or non-primary (NPI) maternal infection. Average transmission rate from mothers to fetuses is estimated around 40% after PI but varies depending on gestational age at maternal CMV infection [3–5]. Risk of long term sequelae is higher if maternal CMV PI occurs in the first trimester of pregnancy or during peri-conceptionnal period [4–7]. It is now established that risk of sequelae among cCMV children is unrelated to maternal type of infection (PI versus NPI) [8–11]. At birth, 13% of cCMV neonates are symptomatic mainly with growth restriction, microcephaly, ventriculomegaly, chorioretinitis, sensorineural hearing loss, hepatitis, thrombocytopenia and a purpuric skin eruption [2, 12].

cCMV infection is a public health issue. However, recommendations and guidelines to manage cCMV infection are scarce, even if an informal International Congenital Cytomegalovirus Recommendation Group, created in 2015, recently published consensus recommendations for prevention and diagnosis of maternal CMV primary infection during pregnancy and for diagnosis and therapy of cCMV in neonates [13]. Currently, in France, national recommendations concerning systematic screening for CMV during pregnancy and at birth are contradictory depending on societies (Academie de Médecine, Collège National des Gynécologues Obstétriciens de France (CNGOF), Haut Conseil de Santé Publique (HCSP)) [14–16] and more widely, management of maternal CMV infection still represents a challenge in most countries.

Diagnosis of maternal CMV PI mainly relies on serology while diagnostic tools for NPI are still to be developed [17, 18]. In this systematic review, we aim to collect biological, clinical, imaging and therapeutic practices currently used to diagnose, monitor and treat CMV infection in the first trimester of pregnancy as this is the period that carries the most important risk of fetal damage. Prognosis value of diagnostic tools, transmission rates and prognosis factors have been extensively reviewed in 2020 [5] and our main goal is now to describe current

practices of maternal CMV infection management in order to provide a support for cost effectiveness analysis.

## Material and methods

### Study design and registration

The systematic review protocol was previously registered in PROSPERO International Prospective Register of systematic reviews (http://www.crd.york.ac.uk/PROSPERO), registration number CRD42019124342. It was conducted and reported in accordance to the Preferred Reporting Items for Systematic Review and Meta-Analysis (PRISMA) 2015 statement [19].

### Eligibility criteria

Eligibility criteria are defined as followed owing to the PICOS definitions (Population–Interventions–Comparators–Outcomes–Studies):

**Type of populations.** Women with CMV PI during the first trimester of pregnancy and fetuses/neonates born to these women. Studies conducted in pregnant women with immunosuppression factors (e.g. autoimmune disease, immunosuppressive treatment, HIV infection) were excluded. Studies focusing on maternal CMV infection diagnosis in pregnant women without mentioning fetal or children outcome were also excluded.

**Type of interventions.** Studies relating to biological, clinical, radiological, therapeutic interventions to diagnose, predict, prevent and treat cCMV were included. Interventions and measures to prevent maternal CMV infections (such as hygiene-based behavioral interventions or hypothetical vaccine), interventions to improve knowledge of CMV infection pathophysiology, and interventions that are no longer available in current practice were excluded.

**Comparator.** A comparator group is irrelevant for this systematic review.

**Type of outcome measures.** Study selection was not performed according to outcome measures criteria.

**Type of studies.** All study designs (randomized controlled trials, controlled trials, observational studies, prospective and retrospective cohort studies. . .) were included, except review articles, letters, case reports and case series with $\leq$ 10 CMV infected women in the first trimester of pregnancy. There was no restriction concerning study duration, study period or date of publication. Only articles written in English or French were included.

### Search strategy

We performed electronically searches on the following databases: MEDLINE, EMBASE, the Cochrane Library, including the Cochrane Central Register of Controlled Trials (CENTRAL), ClinicalTrials.gov, Web of Science until June 1st, 2021. Relevant medical subject heading (MeSH) terms and key words relating to "Cytomegalovirus infection", "congenital" were used as restricting criteria (**S1 Fig**). Grey literature with non-published studies (congress abstracts) were analyzed and we contacted the authors when needed. All references were imported in Zotero (Version 5.0.60) and duplicates were removed.

### Study screening

In a first step, two teams of reviewers (CPD and CVF / CPD and OP) independently screened titles and abstracts to identify relevant studies meeting the pre-specified PICOS inclusion criteria. In a second step, the same reviewers examined full text of selected studies. Discrepancies were solved after discussion with the three reviewers. A flowchart diagram was generated to document the study selection process [20] and the inter-observer agreement between

reviewers was calculated using kappa coefficient [21]. A kappa coefficient higher than 0.6 indicates an acceptable agreement between reviewers [21].

## Data extraction

Data were extracted using a structured Excel sheet. For each eligible article, we extracted the following information if available: country, study design and study period, number of pregnant women with PI in first trimester of pregnancy. For each patient, we collected the following data:

- presence or absence of US abnormalities

- presence or absence of intervention during pregnancy and its result: amniocentesis, maternal viremia, magnetic resonance imaging (MRI), hyperimmunoglobulin (HIG), valaciclovir, fetal blood sampling

- termination of pregnancy (TOP) or alive newborn: infected or not, symptomatic or asymptomatic

    We contacted authors for complementary information in case of missing data.

## Assessment of risk of bias in included studies

Risk of bias was assessed using the Newcastle-Ottawa Scale (NOS) tool for non-randomized cohort studies. Each study was judged on 8 items, categorized into 3 broad groups: selection of the study groups, comparability of the groups, and ascertainment of either the exposure or outcome of interest [22].

    Risk of bias for randomized controlled studies (RCT) was assessed using the Cochrane RoB Tool [23]. Risk of selection, reporting, and other bias were assessed in the Quality Assessment Form Part I. Risk of performance, detection, and attrition bias are assessed using the Quality Assessment Form Part II.

## Data synthesis

We provided a systematic narrative synthesis of the findings from the included studies, structured as type of intervention, study design, intervention content and outcome of interest. Primary outcomes were frequency of interventions for maternal management according to prenatal diagnosis: 1) before amniocentesis, 2) without amniocentesis and 3) after positive amniocentesis and according to 1) presence or 2) absence of ultrasound (US) abnormalities. Biological, radiological and therapeutic interventions in these situations were reported. Secondary outcomes were: TOP or alive newborn, cCMV or not, symptomatic or asymptomatic cCMV, according to radiological findings.

## Statistical analysis

Frequencies of interventions and 95% Confidence Intervals (CI) were calculated with RStudio software (version 1.4.1106). To improve robustness of our results, a sensitivity analysis was carried out. We excluded studies with fair risk of bias and studies for which maternal PI exclusively in first trimester of pregnancy was not confirmed by authors.

# Results

## Study selection

Electronic search from databases yielded 4,134 records after removing duplicates; 4,012 of them were excluded based on title or abstract. We screened 122 studies on full-text review, and

finally included 31 studies in quantitative and qualitative synthesis (**Fig 1**). 88 studies were excluded on full-text because of Population (36 studies), Intervention (30 studies) and Studies (15 studies) exclusion criteria, whereas 7 studies were excluded because they were duplicates. Moreover, two studies were excluded because of high risk of bias and one was excluded because of high risk of overlapping with another study. Details of included studies are described in **Table 1** and excluded studies are described in **S1 Table**.

We distinguished two study populations (**Table 1**):

- pregnant women with primary infection, before or without amniocentesis: 19 studies

- pregnant women with cCMV fetus (positive amniocentesis with or without fetal impairment): patients from 19 studies above and 12 other studies

Kappa coefficient for study screening between OP and CPD was 0.70 and was 0.72 between CVF and CPD.

## Assessment of risk of bias

Risk of bias was assessed using NOS scale for observational studies (**Table 2**). Two studies [54, 55] were excluded from analysis because of high risk of bias in selection domain, comparability domain and outcome domain. **Table 3** shows risk of bias assessment for the 3 randomized controlled trials. Six studies were identifies with fair risk of bias [30, 41, 43, 48, 49, 53]. They were included in main data analysis but were excluded from sensitivity analysis. Moreover, we excluded four studies due to lack of data concerning first trimester PI [27, 38, 39, 46].

## Synthesis of results

**Primary outcomes.** Two populations were analyzed for management of CMV infection during pregnancy. Study population flowchart is described on **Fig 2**. Population 1 consisted of 3,325 pregnant women with CMV PI in first trimester of pregnancy and data on management interventions were extracted from 19 studies out of 31. Population 2 consisted of 1,021 pregnant women with cCMV fetus (positive CMV PCR on amniotic fluid), after a maternal CMV

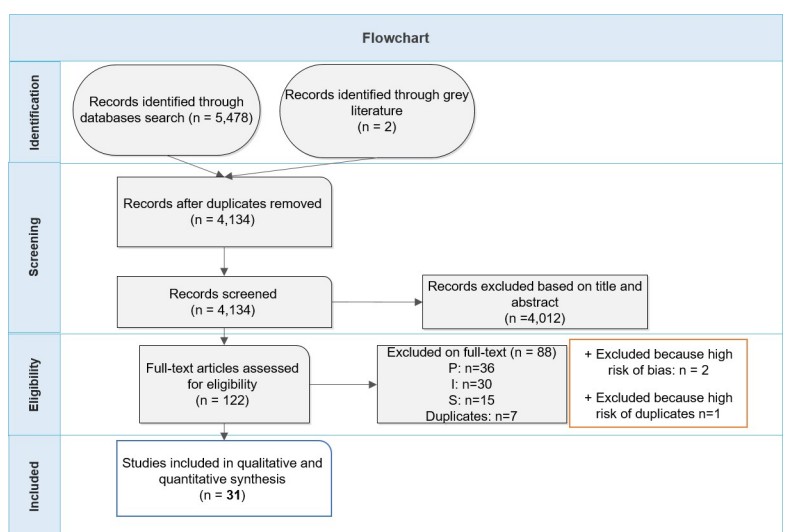

**Fig 1. Flowchart of the inclusion process for study screening.**

**Table 1. Description of included studies.**

| Authors | Country | n women with PI | Interventions of management before or without PD | n women with infected fetus | Interventions of management after positive PD | n TOP |
|---|---|---|---|---|---|---|
| | | Population 1 | | Population 2 | | |
| Blázquez-Gamero 2019 (retrospective) [24] | Spain | 12 | Amniocentesis, US, HIG as prevention | 5 | US, HIG as treatment | 1 |
| Chiaie 2018 (retrospective) [25] | Germany | 20 | Amniocentesis, US, HIG as prevention | 8 | US, HIG as treatment, MRI, FBS | 0 |
| Enders 2017 (retrospective) [26] | Germany | 70 | Amniocentesis, US, HIG as prevention | 30 | US, FBS | 1 |
| Simonazzi 2017 (prospective) [27] | Italy | 182 | Amniocentesis, viremia | 34 | No intervention reported | 17 |
| Delay 2016 (retrospective) [28] | France | No information | No information | 14 | US, MRI, FBS | 8 |
| Leyder 2016 (prospective) [29] | Belgium | No information | No information | 61 | US | 26 |
| Leruez-Ville 2016 (prospective) [30] | France | No information | No information | 40 | US, MRI, valaciclovir as treatment, FBS | 2 |
| Leruez-Ville 2016 (retrospective) [31] | France | No information | No information | 45 | US, valaciclovir as treatment, FBS | 24 |
| Zavattoni 2014 (retrospective) [32] | Italy | No information | No information | 47 | FBS | No information |
| Revello 2014 (randomized controlled trial) [33] | Italy | 57 | Amniocentesis, viremia, HIG as prevention | 15 | No intervention reported | 8 |
| Picone 2013 (retrospective) [4] | France | 72 | Amniocentesis, US | 14 | US, MRI | 7 |
| Visentin 2012 (prospective) [34] | Italy | 591 | Amniocentesis, US | 91 | US, HIG as treatment | 24 |
| Nigro 2012 (case control) [35] | Italy | No information | No information | 51 | US, HIG as treatment | 0 |
| Feldman 2011 (retrospective) [36] | Israel | 152 | Amniocentesis, US | 49 | US, MRI | 37 |
| Revello 2011 (restrospective) [37] | Italy | 371 | Amniocentesis | 104 | FBS | 58 |
| Benoist 2008 (retrospective) [38] | France | No information | No information | 56 | US, MRI, FBS | 29 |
| Guerra 2008 (retrospective) [39] | Italy | 600 | Amniocentesis, US | 15 | No intervention reported | 8 |
| Romanelli 2008 [40] | France | No information | No information | 12 | US, MRI, FBS | 1 |
| Jacquemard 2007 (prospective and retrospective) [41] | France | No information | No information | 45 | US, valaciclovir as treatment, FBS | 21 |
| Nigro 2005 (prospective) [42] | Italy | 57 | Amniocentesis, US, HIG as prevention | 39 | US, HIG as treatment | 0 |
| Lipitz 2002 (prospective) [43] | Israel | No information | No information | 50 | US | 33 |
| Kagan 2019 (prospective) [44] | Germany | 40 | Amniocentesis, viremia, HIG as prevention | 1 | No intervention reported | 0 |
| Lipitz 2019 (prospective) [45] | Israel | No information | No information | 123 | US, MRI | 15 |
| Simonazzi 2019 [46] | Italy | 258 | US | No information | No intervention reported | 3 |
| Shahar-Nissan 2019 (randomized controlled trial) [47] | Israel | 90 | Amniocentesis, US, valaciclovir as prevention | 15 | US, MRI | 6 |
| Hughes 2019 (randomized controlled trial) [48] | USA | 399 | HIG as prevention | No information | No intervention reported | 9 |
| Faure-Bardon 2020 (retrospective cohort study) [49] | France | No information | No information | 62 | US, MRI, FBS | 6 |
| De Santis 2020 (case series) [50] | Italy | 11 | Amniocentesis, US, valaciclovir, viremia | 2 | No intervention reported | 0 |

(*Continued*)

**Table 1.** (Continued)

| Authors | Country | n women with PI | Interventions of management before or without PD | n women with infected fetus | Interventions of management after positive PD | n TOP |
|---|---|---|---|---|---|---|
| | | Population 1 | | Population 2 | | |
| Seidel 2020 (retrospective cohort study) [51] | Germany | 33 | Amniocentesis, HIG as prevention | 2 | No intervention reported | 1 |
| Nigro 2020 (retrospective cohort study) [52] | Italy | 180 | Amniocentesis, US, viremia, HIG as prevention | 73 | MRI | 21 |
| Faure-Bardon 2021 (case control study) [53] | France | 130 | Amniocentesis, valaciclovir as prevention | 27 | No intervention reported | Not reported |

Information was collected for each included study: country; number of women with primary infection (PI) before prenatal diagnosis (population 1) and their management interventions; number of pregnant women with infected fetuses (population 2) and their management interventions; number of terminations of pregnancy (TOP).

PI in first trimester of pregnancy. Data of management interventions were extracted from 30 studies on 31.

*Management of women before prenatal diagnosis (population 1)*. The 3,325 pregnant women came from six countries: France, Germany, Israel, Italy, Spain and USA (**Fig 3B**). Majority of patients were from Italian studies (2,307/3,325: 69%).

Patients were referred for serological criteria for 65% (n = 2,169/3,325) of cases, <1% (n = 3/3,325) for US abnormalities and authors did not report this information for 35% of cases (n = 1,153/3,325).

- amniocentesis and US:
  Amniocentesis was performed in 1,715/2,085 pregnant women (82%) at a median gestational age of 19 WG IQR [18–21] and was not performed for 370/2,085 women (18%). For 1,240/3,325 (37%) women, amniocentesis was not reported (**Fig 2**). Prenatal US examination was performed with a median frequency of 0.75 per month, IQR [0.75–1]. US abnormalities were reported for 157/1,453 cases (11%), whereas in 1,296/1,453 US did not report abnormalities (89%). Focusing on patients with both US and amniocentesis results reported (n = 612 women), US abnormalities were observed in 119/612 cases (19%). Amniocentesis was performed in 97% (115/119) cases if US abnormalities, and performed in 91% (448/493) cases when no US abnormalities was reported. Amniocentesis was significantly more frequently performed in case of abnormal US (p = 0.05) (**Fig 3A**).
  Same results were observed in sensitivity analysis, excluding studies with fair risk of bias and lack of data from the authors (**S2A and S2B Fig**). In sensitivity analysis, we had a population 1 of 1,756 women.

- maternal viremia:
  Maternal viremia was performed for 470/3,325 women (14%) and was reported positive for 113/413 (27%) and negative for 300/413 (73%). Viremia result was not reported for 57/470 women.

- treatments to prevent mother-to-child transmission:
  For 473/3,325 women (14%), HIG was administered; 236 of the 473 (50%) were enrolled in a randomized controlled trial [33, 48]; another 220/3,325 (7%) women received an HIG placebo in this trial. **Fig 3C** shows countries where HIG were administered: USA, Germany, Italy, and Spain; no other countries used HIG. Median dose used for HIG as prevention was 100 IU/kg IQR [100–200] (**Fig 3D**), median gestational age at first administration was 16 WG IQR [14–16] and the median number of administered doses was 5 IQR [2–5].

**Table 2. Evaluation of risk of bias for observational studies (NOS scale).**

| Study | Selection | | | | Comparability [5] | Outcome | | | Quality |
|---|---|---|---|---|---|---|---|---|---|
| | Representativeness of the exposure (intervention) cohort [1] | Selection of the nonexposed cohort [2] | Ascertainment of exposure [3] | Incident disease [4] | | Assessment of outcome [6] | Lenght of follow up [7] | Adequacy of follow up [8] | |
| Blázquez-Gamero, 2019 [24] | A | B | A | A | B | B | A | A | **Good** |
| Chiaie, 2018 [25] | A | B | A | A | B | B | A | B | **Good** |
| Beloosesky, 2017 [54] | D | C | D | A | C | B | B | D | **Poor** |
| Enders, 2017 [26] | A | A | A | A | A | B | A | B | **Good** |
| Simonazzi, 2017 [27] | C | A | A | A | A | B | A | A | **Good** |
| Delay, 2016 [28] | C | A | A | A | A | B | A | A | **Good** |
| Leyder, 2016 [29] | C | A | A | A | A | B | A | B | **Good** |
| Cannie, 2016 [55] | C | C | A | B | C | C | A | C | **Poor** |
| Leruez-Ville, 2016 [30] | C | B | A | A | A | B | A | B | **Fair** |
| Leruez-Ville, 2016 [31] | C | B | A | A | A | B | A | B | **Good** |
| Zavattoni, 2014 [32] | C | A | A | A | A | B | A | A | **Good** |
| Picone, 2013 [4] | A | B | A | A | A | B | A | A | **Good** |
| Visentin, 2012 [34] | A | A | A | A | A | B | A | A | **Good** |
| Nigro, 2012 [35] | C | A | A | A | A | B | A | A | **Good** |
| Feldman, 2011 [36] | A | A | A | A | A | B | A | A | **Good** |
| Revello, 2011 [37] | A | A | A | A | A | B | A | C | **Good** |
| Benoist, 2008 [38] | B | B | A | A | A | B | A | A | **Good** |
| Guerra, 2008 [39] | A | A | A | A | A | A | A | B | **Good** |
| Romanelli, 2008 [40] | C | A | A | A | A | B | A | A | **Good** |
| Jacquemard, 2007 [41] | C | B | A | A | A | B | A | A | **Fair** |
| Nigro, 2005 [42] | B | A | A | A | B | B | A | C | **Good** |
| Lipitz, 2002 [43] | C | B | A | A | A | B | A | A | **Fair** |
| Kagan, 2019 [44] | A | B | A | A | B | B | A | A | **Good** |
| Lipitz, 2019 [45] | C | A | A | A | A | B | A | A | **Good** |
| Simonazzi, 2019 [46] | A | A | A | A | A | B | A | A | **Good** |
| Faure-Bardon, 2020 [**49**] | C | B | A | B | A | B | A | B | **Fair** |
| De Santis, 2020 [**50**] | A | B | A | A | A | B | A | B | **Good** |
| Seidel, 2020 [**51**] | A | B | B | A | B | B | A | B | **Good** |
| Nigro, 2020 [**52**] | A | A | A | A | A | A | A | B | **Good** |
| Faure-Bardon, 2021 [**53**] | A | B | A | A | A | B | B | D | **Fair** |

[1] A: Truly representative of the average first trimester infected pregnant women; B: Somewhat representative of the average first trimester infected pregnant women; C: Selected group; D: No description of the derivation of the cohort.

[2] A: Drawn from the same community as the exposed cohort (concurrent controls); B: Drawn from a different source (historical controls); C: No description of the derivation of the non exposed cohort.

[3] A: Secure record (e.g., hospital records); B: Structured interview; C: Written self report; D: No description.

[4] Demonstration that outcome of interest was not present at start of study: A: Yes; B: No.

[5] Comparability of cohorts on the basis of the design or analysis: A: Study controls for age, sex and marital status; B: Study controls for any additional other factor; C: not carried out or not reported.

[6] A: Independent blind assessment; B: Record linkage; C: Self report; D: No description.

[7] Was follow-up long enough for outcomes to occur? A: Yes; B: No.

[8] A: Complete follow up, all subject accounted for; B: Subjects lost to follow up unlikely to introduce bias; number lost less than or equal to 20% or description of those lost suggested no different from those followed; C: Follow up rate less than 80% and no description of those lost; D: No statement.

Sensitivity analysis for preventive HIG shows that Italy and Germany are now the countries with the most HIG use. The dose administered is 200 IU/kg for 80% of patients (**S2C and S2D Fig**).

Table 3. Evaluation of risk of bias for randomized controlled trials (RCT) with Cochrane RoB tool.

| Authors | Selection bias—random sequence generation | Selection bias—allocation concealment | Reporting bias—selective reporting | Other bias | Performance bias—blinding | Detection bias—blinding | Attrition bias—incomplete outcome data | Risk of bias |
|---|---|---|---|---|---|---|---|---|
| Revello, 2014 [33] | low | low | low | low | low | low | low | **low** |
| Shahar-Nissan, 2019 [47] | low | low | low | low | low | low | low | **low** |
| Hughes, 2019 [48] | unclear | unclear | low | low | unclear | unclear | low | **fair** |

Valaciclovir P.O. 8g per day as prevention of mother-to-child CMV transmission was administered to 121/3,325 patients (4%), including 45 in a randomized controlled trial; 45/3,325 other patients (1%) received placebo. Median gestational age at first administration was 12.3 WG IQR [11.4–13.0], treatment was ended at 22 WG.

*Management of population with fetal infection proven by a positive CMV PCR in amniotic fluid (population 2).* In this review, 1,021 pregnant women had infected fetuses. Six countries were represented: Belgium, France, Germany, Israel, Italy and Spain (**Fig 4B**).

Patients were referred for serological criteria for 54% (n = 549/1,021) of cases, 2% (n = 20/1,021) for US abnormalities and authors did not report this information for 44% of cases (n = 452/1,021).

- *US and MRI*:

The mean frequency of US was 1.5 examinations per month IQR [1–2]. US abnormalities were reported in 360/832 (43%) fetuses (**Fig 4A**). MRI was performed for 390/890 pregnancies (44%), and showed abnormalities in 75/224 (33%). MRI result was not reported for 166/390 pregnancies (43%). Median gestational age for MRI was 32 WG IQR [31–32]. MRI was performed in 47% (224/472) of patients with normal US, whereas MRI was performed in

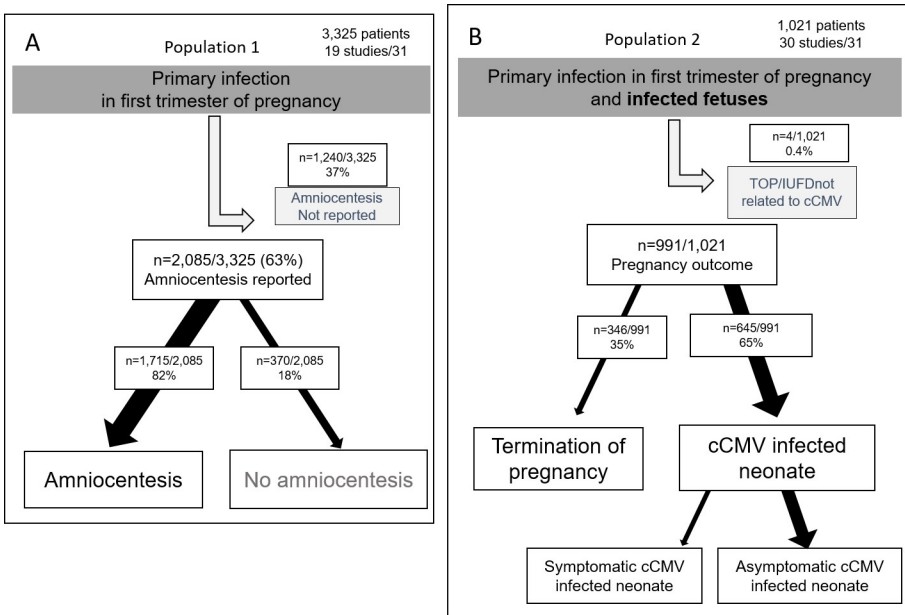

**Fig 2. Population flowchart.** Population was divided into two categories of population: A) population 1, with pregnant women before amniocentesis and B) population 2, after positive amniocentesis. The thickness of the arrows is proportional to the number of patients.

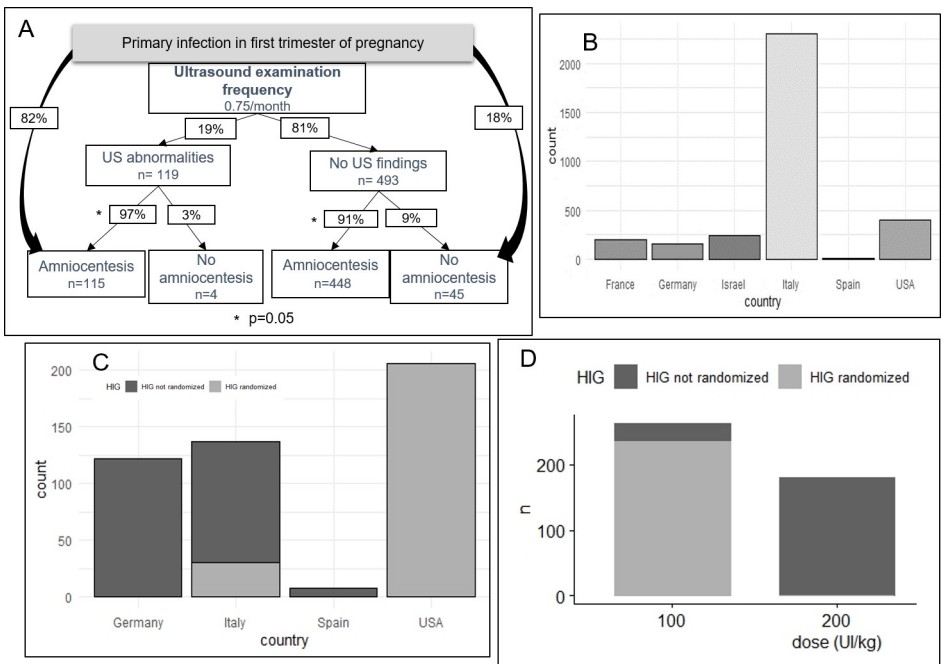

**Fig 3. Management for population 1.** Population 1 flowchart with radiological interventions. Amniocentesis is described according to US abnormalities or not. Patients receiving HIG in RCT are colored in middle grey and patients receiving HIG in observational studies are colored in dark grey. A. Repartition of population 1 according to their country of study. B. Repartition of patients receiving HIG for preventive treatment according to their country. C. Repartition of patients receiving HIG for preventive treatment according to dose of HIG: 100 IU/kg or 200 IU/kg. US: ultrasound; HIG: hyperimmunoglobulin; RCT: randomized controlled trial.

44% (158/360) of patients with US abnormalities (not significative) (**Fig 4A**).
**Fig 4C** shows the comparison of US and MRI results. In case of US abnormalities, MRI abnormalities were observed in 53% (60/114) of cases. In cases with no US abnormalities, MRI abnormalities were observed in 14% (15/110) (p<0.001).
Sensitivity analysis gave similar results with main analysis (**S3 Fig**).

- *fetal blood sampling*:
  Fetal blood sampling was performed in 246/1,021 women (24%) at a median gestational age of 28 WG IQR [26–30]. Fetal viral load and fetal platelet counts were performed in respectively 96% (236/246) and 85% (210/246) of these women. FBS was performed in 40% of patients (144/360) with US abnormalities and was performed in 14% (66/472) patients with normal US (p<0.001).

- *curative therapies*:
  HIG was administered in 92/1,021 women (9%) at a dose of 200 IU/kg. The median gestational age at first administration was 22 WG IQR [21–24] and the median number of doses was 1 IQR [1–2].
  Valaciclovir 8g per day orally was used in 77/1,021 women (8%), at a median gestational age of 28 WG IQR [25–32] until the end of pregnancy.

**Secondary outcomes.** **Fig 4D** shows outcome of pregnancy for women with infected cCMV fetuses, according to imaging findings. Even in case of normal US examination, TOP was performed in 29% of cases (104/362) and neonates were symptomatic in 13% of cases (48/362). Abnormal imaging findings were associated with a higher proportion of TOP and a

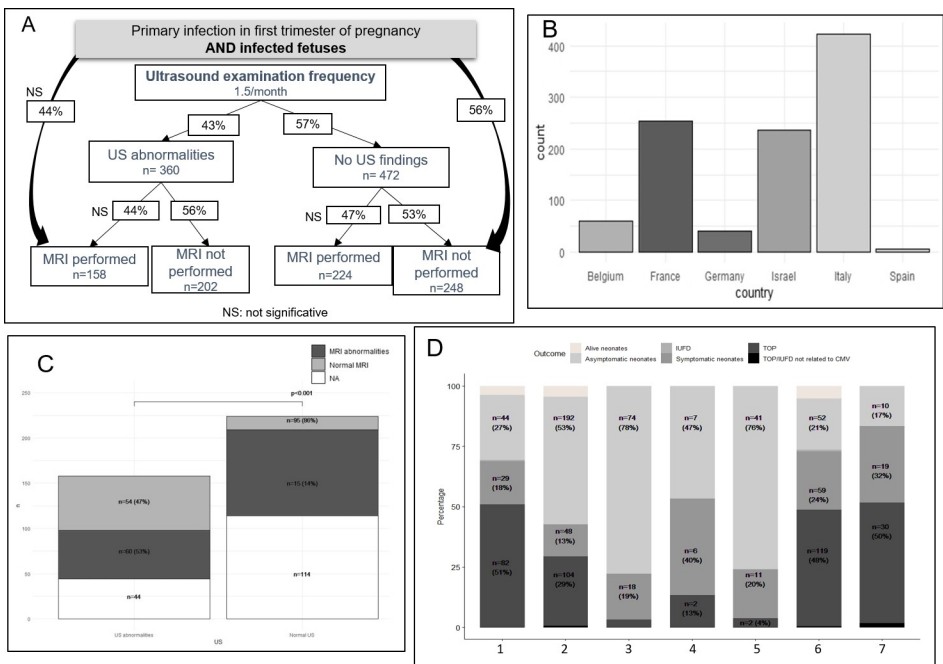

**Fig 4. Management for population 2 (women with infected fetuses).** A. Population 2 flowchart with radiological interventions (US and MRI). MRI performing is described according to US abnormalities or not. B. Repartition of population 2 according to their country of study. C. MRI findings according to US abnormalities. D. Pregnancy outcome (alive neonate–symptoms not reported, asymptomatic neonate, IUFD, symptomatic neonate, TOP, TOP/IUFD not related to cCMV) according to radiological findings (1 to 7). 1: presence of US findings? Data not reported; 2: Normal US; 3: Normal US and normal MRI; 4: Normal US and MRI abnormalities; 5: US abnormalities and normal MRI; 6: US abnormalities; 7: US abnormalities and MRI abnormalities. NA: data not reported; US: ultrasound; MRI: magnetic resonance imaging; HIG: hyperimmunoglobulin; RCT: randomized controlled trial; IUFD: intra uterine fetal death.

lower proportion of asymptomatic infected neonates. Percentage of symptomatic infected neonates at birth (as considered by authors) was statistically higher in case of US abnormalities and in case of US and MRI abnormalities (p<0.001).

Sensitivity analysis did not change results for pregnancy outcome according to radiological findings (**S4A Fig**). Performing a second sensitivity analysis without Italian nor Israeli women (n = 180) (**S4B Fig**), 23% (19/81) of TOP were reported in case of normal US.

## Discussion

### Main findings

In this review of management practices, in pregnant women with CMV PI before or without amniocentesis, US examination frequency was a mean of 0.75/month, amniocentesis was performed in 82% and prophylaxis was prescribed, with HIG or valaciclovir in 14% and 4%, respectively. Amniocentesis was significantly more frequently performed in case of US abnormalities. In case of cCMV, US was performed 1.5/month, MRI was performed in 44%, FBS in 24% and therapy with HIG or valaciclovir in respectively 9% and 8% of patients. This shows that diagnosis of cCMV infection in fetuses with amniocentesis allows for more intensive work-up including US, MRI and FBS. The overall rate of US abnormalities might be overestimated for fetuses with cCMV because some were diagnosed following the detection of abnormalities during routine screening ultrasound. This is probably not the case because majority of patients was addressed because of serological criteria.

For sensitivity analysis, studies with fair risk of bias were excluded as well as studies in which authors did not confirm that patients had PI in first trimester of pregnancy. Sensitivity analysis confirmed our results, which emphasizes our management description.

Our results are globally comparable with previously published studies about management process of pregnant women with CMV, including practice surveys [56] and literature reviews [57–60]. While amniocentesis and US examination are widely performed, approaches to FBS, MRI, and therapies are more diverse. One common limitation in systematic review is the potential presence of publication bias which could in our case overestimate these interventions compared to real life practices. In 2016, Carrara *et al.* [56] published management practices after CMV PI in pregnant women in France, from a survey of gynecologists' practices. Amniocentesis was performed in only 57% to 62% of cases after a PI in the first trimester of pregnancy without US abnormalities, compared to 91% in our study. On the contrary, MRI was more frequently prescribed, between 70% if normal US and more than 80% if abnormal US, compared to 44% to 47% in our study. For fetal blood sampling, we had similar results to Carrara *et al.*, who had 12% of FBS in patients with normal US and 44% of FBS if US abnormalities.

Regarding outcomes for infected fetuses, TOP was reported in 29% of cases and 18% of neonates were symptomatic when "normal US". Non-surprisingly, abnormal imaging findings were associated with a higher proportion of TOP and a lower proportion of asymptomatic infected neonates. We classified newborns as symptomatic based on symptoms ranging from "asymptomatic congenital infection with isolated sensorineural hearing loss" to "severely symptomatic congenital infection" as defined by Rawlinson *et al.* [13]. The more US/MRI abnormalities were observed, the more TOP occurred, as well as asymptomatic infected neonates decreased. The proportion of symptomatic infected neonates was higher in case of US +/- MRI abnormalities, both in main and in sensitivity analysis. As Italy and Israel represented the majority of patients studied in these two analyses (**Fig 4B** and **S3B Fig**), we performed a second sensitivity analysis excluding Italian and Israeli patients from analysis in order to avoid bias in pregnancy outcome, because legislation on TOP is more restricted in Italy and Israel compared with other countries [61, 62]. This second sensitivity analysis reported similar results with 23% TOP performed and 9% symptomatic neonates in case of normal US. In summary, the proportion of TOP in case of normal US without MRI performed was between 23% and 31%. This high proportion of TOP without abnormal imaging is of ethical concern. According to Carrara *et al.* [56], TOP was unacceptable for 94% of practitioners in case of normal US and unacceptable for 78% in case of minor US abnormalities. However, TOP was acceptable for 90% of them in case of major US abnormalities and normal MRI. Hui and Wood [60] reviewed perinatal outcome after CMV PI during first trimester of pregnancy in published literature. TOP was performed for 7% of infected fetuses, whereas we identified 35% of TOP in our study.

Our study is original and objective as this is a systematic review conducted to provide an overview of current practice in management of maternal and congenital CMV during pregnancy. To the best of our knowledge, this has never been performed before. Strengths of our study include robustness and quality of methodology of the systematic review, according to PRISMA statement. Our results reflect management of the infection in real life, once the diagnosis of maternal CMV PI is made with serological tools is made (seroconversion and/or presence of CMV IgM with low CMV IgG avidity).

## Limitations of our study

The main limitation of our study is heterogeneity in study designs (observational, prospective, and retrospective, RCT, case-control), population studied and representativeness of countries. Indeed, majority of data is from Italy, followed by France and Israel because these countries

are particularly involved in CMV management. However, we believe that their practices may be representative and followed in countries with similar epidemiology and health care resources. Observational studies were more represented than RCT, with an inherent higher risk of heterogeneity in observational studies rather than RCT. Although we had a higher proportion of observational studies, these studies are probably more representative of real life practices, which is the objective of our systematic review.

We noticed heterogeneity in management according to studies included. That emphasizes the interest of our systematic review, to obtain a global overview of current management practices for maternal and congenital CMV infection during pregnancy. Heterogeneity is also due to over-representation of Italian patients, particularly among population 1. In two Italian studies with a high number of patients [39, 46], we questioned authors about exhaustiveness of first trimester maternal CMV PI. In absence of answer from the authors, we decided to exclude these studies in sensitivity analysis.

Our systematic review is a snapshot of current practices in management of maternal and cCMV infection during pregnancy after PI occurring in the first trimester of pregnancy. Of course these practices are likely to evolve, in particular following the RCT demonstrating a preventive effect of valaciclovir on mother to fetus CMV transmission [47]. These interventions should not make us forget that prevention for CMV infection relies mainly on hygiene measures for pregnant women and future fathers.

## Conclusions

Our systematic review draws the picture of management interventions in pregnant women after a CMV PI occurring in first trimester of pregnancy, according to published and non-published literature. Questions about management guidelines remain and this systematic review will provide a support for cost-effectiveness analysis.

## Supporting information

**S1 Fig. MEDLINE database retrieval strategy.**
(TIF)

**S2 Fig. Management for population 1 in sensitivity analysis.** A. Repartition of population 1 according to their country of study. B. Population 1 flowchart with radiological interventions (US and or RMI). Amniocentesis is described according to US abnormalities or not. C. Repartition of patients receiving HIG for preventive treatment according to their country. D. Repartition of patients receiving HIG for preventive treatment according to dose of HIG: 100 UI/kg or 200 UI/kg. Patients receiving HIG in RCT are colored in middle grey and patients receiving HIG in observational studies are colored in dark grey. US: ultrasound; HIG: hyperimmunoglobulin; RCT: randomized controlled trial.
(TIF)

**S3 Fig. Management for population 2 in sensitivity analysis.** A. Population 2 flowchart with radiological interventions (US and MRI). MRI performing is described according to US abnormalities or not. B. Repartition of population 2 according to their country of study. C. MRI findings according to US abnormalities. NA: data not reported; US: ultrasound; MRI: magnetic resonance imaging.
(TIF)

**S4 Fig. Outcome of pregnancy for population 2 in sensitivity analysis.** Pregnancy outcome (alive neonate–symptoms not reported, asymptomatic neonate, IUFD, symptomatic neonate, TOP, TOP/IUFD not related to cCMV) is reported according to radiological findings (1 to 7))

in sensitivity analysis for population with infected fetuses. A. First sensitivity analysis: without studies with fair risk of bias B. Second sensitivity analysis: without studies with fair risk of bias and without studies with nor Italian patients nor Israeli patients. 1: presence of US findings? Data not reported; 2: Normal US; 3: Normal US and normal MRI; 4: Normal US and MRI abnormalities; 5: US abnormalities and normal MRI; 6: US abnormalities; 7: US abnormalities and MRI abnormalities. US: ultrasound; MRI: magnetic resonance imaging; TOP: termination of pregnancy; IUFD: intra uterine fetal death.
(TIF)

**S1 Table. Description of excluded studies on full text and reasons of exclusion.**
(DOCX)

**S1 Checklist.**
(DOC)

## Author Contributions

**Conceptualization:** Claire Périllaud-Dubois, Drifa Belhadi, Cédric Laouénan, Olivier Picone, Christelle Vauloup-Fellous.

**Investigation:** Claire Périllaud-Dubois.

**Methodology:** Claire Périllaud-Dubois, Drifa Belhadi, Cédric Laouénan, Laurent Mandelbrot, Olivier Picone, Christelle Vauloup-Fellous.

**Software:** Claire Périllaud-Dubois, Drifa Belhadi.

**Supervision:** Drifa Belhadi, Cédric Laouénan, Laurent Mandelbrot, Olivier Picone, Christelle Vauloup-Fellous.

**Validation:** Drifa Belhadi, Cédric Laouénan, Laurent Mandelbrot, Olivier Picone, Christelle Vauloup-Fellous.

**Writing – original draft:** Claire Périllaud-Dubois.

**Writing – review & editing:** Drifa Belhadi, Cédric Laouénan, Laurent Mandelbrot, Olivier Picone, Christelle Vauloup-Fellous.

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
