## [Decision Letter · Decision Letter 0]

1 Oct 2021

PONE-D-21-17252

Current practices of management of maternal and congenital Cytomegalovirus infection during pregnancy after a maternal primary infection occurring in first trimester of pregnancy: systematic review

PLOS ONE

Dear Dr. Périllaud-Dubois,

Thank you for submitting your manuscript to PLOS ONE. After careful consideration, we feel that it has merit but does not fully meet PLOS ONE’s publication criteria as it currently stands. Therefore, we invite you to submit a revised version of the manuscript that addresses the points raised during the review process.

We look forward to receiving your revised manuscript.

Kind regards,

Linglin Xie

Academic Editor

PLOS ONE

Additional Editor Comments (if provided):

Reviewers' comments:

Reviewer's Responses to Questions

**Comments to the Author**

1. Is the manuscript technically sound, and do the data support the conclusions?

Reviewer #1: Yes

Reviewer #2: Yes

2. Has the statistical analysis been performed appropriately and rigorously? 

Reviewer #1: I Don't Know

Reviewer #2: Yes

3. Have the authors made all data underlying the findings in their manuscript fully available?

Reviewer #1: Yes

Reviewer #2: Yes

4. Is the manuscript presented in an intelligible fashion and written in standard English?

Reviewer #1: Yes

Reviewer #2: Yes

5. Review Comments to the Author

Reviewer #1: do you have any reference(s) for the practice and legislation regarding TOP in Israel and Italy? it would be very helpful if you can provide some.

PLease pay attention to the references. some are incompletely provided, i. e; 14, 16, 21, 22, 23....etc.

Reviewer #2: The article has covered published data from seven countries. The data compiled is from different studies published in these countries till dates. The statistical analysis is remarkable. The title "Current practices of management of maternal and congenital Cytomegalovirus infection during pregnancy after a maternal primary infection occurring in first trimester of pregnancy: systematic review: The current review included 31 out of 4134 studies after excluding large number of studies. The objective was to extract information about biological. Clinical, imaging and therapeutic practices. However, looking into the data it appears that majority of data is from Italy followed by France. The review could not provide any conclusive new information that can be adopted by a practitioner. The data is variable and scattered. It does not specifically arrive at any specific conclusion about its general objectives.

6. PLOS authors have the option to publish the peer review history of their article (what does this mean?). If published, this will include your full peer review and any attached files.

Reviewer #1: No

Reviewer #2: **Yes: **Ramesh K Chandolia

---

## [Author Response · Author response to Decision Letter 0]

24 Oct 2021

Reviewer #1: do you have any reference(s) for the practice and legislation regarding TOP in Israel and Italy? it would be very helpful if you can provide some.

Please pay attention to the references. some are incompletely provided, i. e; 14, 16, 21, 22, 23... etc.

We thank the first reviewer for his interesting suggestion. We added two legislation references (line 361):

61: Italian legislation concerning termination of pregnancy (articles 6 and 7): Leggi Normativa: Norme per la tutela sociale della maternita' e sull'interruzione volontaria della gravidanza. (G.U. Serie Pregressa, n. 140 del 22 maggio 1978) Articles 6,7. Available at https://www.altalex.com/documents/leggi/2008/05/09/tutela-sociale-della-maternita-ed-interruzione-volontaria-della-gravidanza

62: Israeli legislation concerning late termination of pregnancy: Ministry of Health Director Circular 23/07 from 19.12.2007 regarding late termination of pregnancy committees. Available at ttps://www.kolzchut.org.il/en/Termination_of_Pregnancy

As required, we completed references that were incomplete.

Reviewer #2: The article has covered published data from seven countries. The data compiled is from different studies published in these countries till dates. The statistical analysis is remarkable. The title "Current practices of management of maternal and congenital Cytomegalovirus infection during pregnancy after a maternal primary infection occurring in first trimester of pregnancy: systematic review: The current review included 31 out of 4134 studies after excluding large number of studies. The objective was to extract information about biological. Clinical, imaging and therapeutic practices. However, looking into the data it appears that majority of data is from Italy followed by France. The review could not provide any conclusive new information that can be adopted by a practitioner. The data is variable and scattered. It does not specifically arrive at any specific conclusion about its general objectives.

We are grateful for the second reviewer for careful reading of our manuscript. Indeed, majority of data is from Italy, followed by France and Israel because these countries are particularly involved in CMV management. We agree that our review could not provide any conclusive new information because the data are variable and scattered. However, we believe that their practices may be representative and followed in countries with similar epidemiology and health care resources. 

We completed our discussion considering his remarks (lines 379-382).

---

## [Decision Letter · Decision Letter 1]

23 Nov 2021

Current practices of management of maternal and congenital Cytomegalovirus infection during pregnancy after a maternal primary infection occurring in first trimester of pregnancy: systematic review

PONE-D-21-17252R1

Dear Dr. Périllaud-Dubois

We’re pleased to inform you that your manuscript has been judged scientifically suitable for publication and will be formally accepted for publication once it meets all outstanding technical requirements.

Kind regards,

Linglin Xie

Academic Editor

PLOS ONE

Additional Editor Comments (optional):

Reviewers' comments:

Reviewer's Responses to Questions

**Comments to the Author**

1. If the authors have adequately addressed your comments raised in a previous round of review and you feel that this manuscript is now acceptable for publication, you may indicate that here to bypass the “Comments to the Author” section, enter your conflict of interest statement in the “Confidential to Editor” section, and submit your "Accept" recommendation.

Reviewer #1: All comments have been addressed

2. Is the manuscript technically sound, and do the data support the conclusions?

Reviewer #1: Yes

3. Has the statistical analysis been performed appropriately and rigorously? 

Reviewer #1: I Don't Know

4. Have the authors made all data underlying the findings in their manuscript fully available?

Reviewer #1: Yes

5. Is the manuscript presented in an intelligible fashion and written in standard English?

Reviewer #1: (No Response)

6. Review Comments to the Author

Reviewer #1: (No Response)

7. PLOS authors have the option to publish the peer review history of their article (what does this mean?). If published, this will include your full peer review and any attached files.

Reviewer #1: **Yes: **Naser Al-Husban

---

## [Editor Report · Acceptance letter]

26 Nov 2021

PONE-D-21-17252R1 

Current practices of management of maternal and congenital Cytomegalovirus infection during pregnancy after a maternal primary infection occurring in first trimester of pregnancy: systematic review 

Dear Dr. Périllaud-Dubois:

I'm pleased to inform you that your manuscript has been deemed suitable for publication in PLOS ONE. Congratulations! Your manuscript is now with our production department. 

Kind regards, 

on behalf of

Dr. Linglin Xie 

Academic Editor

PLOS ONE